# Programmable base editing of zebrafish genome using a modified CRISPR-Cas9 system

Yihan Zhang[1,2], Wei Qin[1], Xiaochan Lu[1], Jason Xu[2], Haigen Huang[2], Haipeng Bai[1], Song Li[1] & Shuo Lin[1,2]

Precise genetic modifications in model animals are essential for biomedical research. Here, we report a programmable "base editing" system to induce precise base conversion with high efficiency in zebrafish. Using cytidine deaminase fused to Cas9 nickase, up to 28% of site-specific single-base mutations are achieved in multiple gene loci. In addition, an engineered Cas9-VQR variant with 5'-NGA PAM specificities is used to induce base conversion in zebrafish. This shows that Cas9 variants can be used to expand the utility of this technology. Collectively, the targeted base editing system represents a strategy for precise and effective genome editing in zebrafish.

[1] Laboratory of Chemical Genomics, School of Chemical Biology and Biotechnology, Peking University Shenzhen Graduate School, Shenzhen, 518055, China. [2] Department of Molecular, Cell and Developmental Biology, University of California, Los Angeles, 90095 CA, USA. Yihan Zhang and Wei Qin contributed equally to this work. Correspondence and requests for materials should be addressed to S.L. (email: shuolin@ucla.edu)

The clustered regularly interspaced short palindromic repeat (CRISPR) system has been developed as a versatile genome editing tool in various organisms, including zebrafish[1, 2]. The Cas9 protein, guided by guide RNA (gRNA), binds to the target DNA site on the genome and works as a nuclease to induce double-strand breaks (DSBs)[1]. As a natural cellular response, the DSBs are mainly repaired through the non-homologous end joining pathway. This process can generate random deletions or insertions. In addition, specific modifications, such as the substitutions of single bases, and the insertion of longer sequences like loxP elements, can be introduced into the genome with the presence of the homologous donor template through homology-directed repair (HDR)[3–5]. Genome-wide association study coupled with the next-generation sequencing has identified a growing number of candidate genes with single-base mutations associated with human diseases. Inevitably, efficient methods are required to validate the causal mutations responsible for disease phenotypes[6]. The most desirable approach is to introduce the human genetic mutations in model organisms by knock-in using the CRISPR-mediated HDR. Unfortunately, the efficiency of this donor-dependent HDR is low, which restricts the utility of this method[7].

Recently, a technology called "base editing" (BE) was reported, which enables direct and irreversible conversion of one targeted base to another in cultured mammalian cells in a programmable manner without the need for a DSB[8]. In this system, a cytidine deaminase was fused to the N terminus of a Cas9 nickase (nCas9), which mediates the direct conversion of C→T (or G→A) in human cells. The optimal deamination sites for this system are located in a 5 bp window on the CRIPSR–Cas9 target site, −17 to −13 upstream of the PAM sequence. Cas9 nickase maintains its activity to bind DNA with a gRNA and can only cut the non-edited strand, preventing DSBs. By nicking the non-edited DNA strand, both the newly synthesized DNA and damaged DNA are stimulated to resolve the U:G mismatch into T:A, improving the base conversion efficiency. In order to prevent U:G to C:G reversion, a UDG inhibitor (UGI) from Bacillus subtilis bacteriophage PBS1 was fused to the C terminus of nCas9. With this design, it is reported that the BE system can achieve permanent correction of 15–75% of total cellular DNA with minimal (typically ≤1%) indel formation[8]. Conceptually, the BE system should have great potential applications in gene editing by introducing single-base changes to correct or mimic mutations of human genetic disorders in model animals. To date, this system

has been reported to work in mouse and several crops[9–11]. However, it has not been tested if this system will work in zebrafish.

Here, we demonstrate that this BE system can achieve base substitution at efficiency between 9.25 and 28.57% with very low indel formation in zebrafish. To enrich the toolbox of this BE system, we also replace the Cas9 nickase with VQR variant nickase, which recognizes the 5′-NGA PAM. Sequencing results indicate that this BE-VQR system also induce efficient base substitution in a targeted manner. Overall, we demonstrate that the deaminase-Cas9 tool of "base editing" provides a simple and efficient method for introducing single-base changes in zebrafish.

## Results

**BE system can induce base conversion in zebrafish.** To explore whether the BE (rAPOBEC1-XTEN-nCas9-UGI)–gRNA nuclease complex can catalyze site-specific base conversion of zebrafish genome *in vivo*, we first replaced the Cas9 sequence in the original plasmid with a zebrafish codon-optimized version[12]. To assess the feasibility of its use in zebrafish, four target sites from three genes (*twist2, gdf6, and ntl*) were selected (Fig. 1a). After injecting BE messenger RNA (mRNA) and related gRNA into one-cell embryos of zebrafish, six embryos at 48 h post fertilization (hpf) were randomly selected and mixed together for genomic DNA extraction. PCR amplification of the region covering the target site was performed, and the products were used for both sequencing and T-A cloning. Sequencing results of PCR products revealed overlapping signal peaks at the position where the targeted cytidine was located. Outside the targeted cytidine site, no other significant overlapping signal peaks were observed in this region. These results were further confirmed by sequencing individual clones. For *twist2*–1, 4 of the 20 colonies carried C→T conversions, 1 carried a C→A conversion and the remaining 15 maintained wild-type sequence (Fig. 1b). For *gdf6*, 4 of the 20 colonies showed C→A conversions and 1 showed C→T conversion (Fig. 1c). Finally for *ntl*, C-T conversion was observed in 4 of the 24 colonies (Fig. 1d). These types of base conversions were transmitted through germline and identified in embryos of F1 generation. PCR and sequencing analysis showed that 6 out of 16 F0 injected fish transmitted the expected base change to F1 for *twist2*–1 and 2 out of 7 for *ntl*.

The p.E75K mutation in *twist2* was previously reported to be the causative mutation of ablepharon macrostomia syndrome

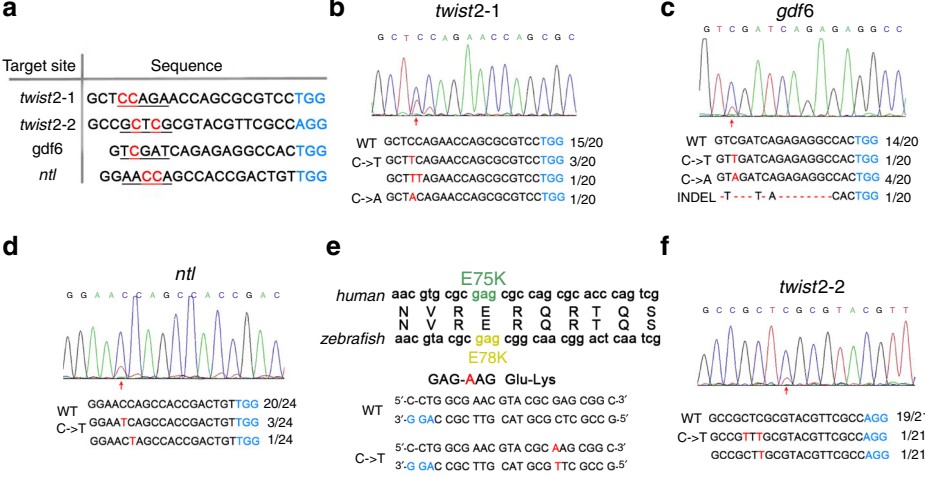

**Fig. 1** Single-base editing in zebrafish. **a** The target site sequences. Target sequence (*black*), PAM region (*blue*), target sites (*red*), and target windows (*underlined*) are indicated. **b–d**, **f** Sequencing results at the *twist2–1, gdf6, ntl,* and *twist2–2* targets. **e** The diagram of mutation of human AMS. *Red arrows* indicate the overlapped peaks. The substituted bases are marked in *red*. *Red dashes* represent the deleted bases in the sequence

(AMS)[13]. Notably, the conversion of C-T in *twist2–2* induced a p.E78K amino-acid change, precisely mimicking the p.E75K mutation found in human (Fig. 1e). Sequencing results showed that 2 out of 21 colonies carried C→T conversion (Fig. 1f). The p.E78K mutation in *twist2–2* was transmitted to the next generation with efficiency of 7.7% (2/26). These results indicate that a zebrafish AMS model precisely mimicking the human mutation can be achieved, suggesting the potential of this system to develop animal models for human disease. Taken together, these data show that base-edited zebrafish can be efficiently generated using this BE-gRNA system.

**Base conversion at the *tyr* locus to mimic OCA mutation.** *tyr* gene has been validated as the most common causal gene for human ocular albinism (OA) and oculocutaneous albinism (OCA). Many point mutations on *tyr* locus have been verified to cause loss-of-function of the protein, resulting in deficiency in pigment formation. Recently, the mutation p.P301L in *tyr* has been found in OCA patients (Fig. 2a)[14]. This suggests that the proline at the position 301 may play an important role in the function of the protein. To test this hypothesis, this amino acid needs to be specifically mutated without changing other sequences of this gene. In order to observe the phenotype directly in F0 fish, we co-injected the *tyr* gRNA/BE mRNA into *tyr* heterozygous zebrafish embryos and then assessed the base conversion. Because of the codon preference of different species, we could not convert proline to leucine at position 302 in zebrafish. Instead, proline was converted to serine, threonine, or alanine (Fig. 2b, c). The injected embryos showed the phenotype of pigment deficiency in eyes (Fig. 2d). This could reflect the importance of this amino acid to keep *tyr* function, but it does not exclude the possibility that random indels caused the phenotype. Analysis of germline transmission in F1 embryos identified CCC→TCC and CCC→TTC (4/25). Screening for other types of base changes is needed to determine which amino-acid change will result in OCA phenotype.

**BE system using Cas9-VQR variant.** To date, many artificial Cas9 variants with altered PAM specificities have been developed to address the limitation of Cas9 PAM requirment[15]. Among them, Cas9-VQR variant which recognizes 5′-NGA PAM has been shown previously to induce indels in zebrafish and *Caenorhabditis elegans*[16, 17]. We replaced the nCas9 in BE/gRNA system with an nCas9-VQR variant to create a BE-VQR fusion protein. Then three gene loci, *twist2*, *tial1*, and *urod* were tested to see if BE-VQR could induce base conversion at four VQR target

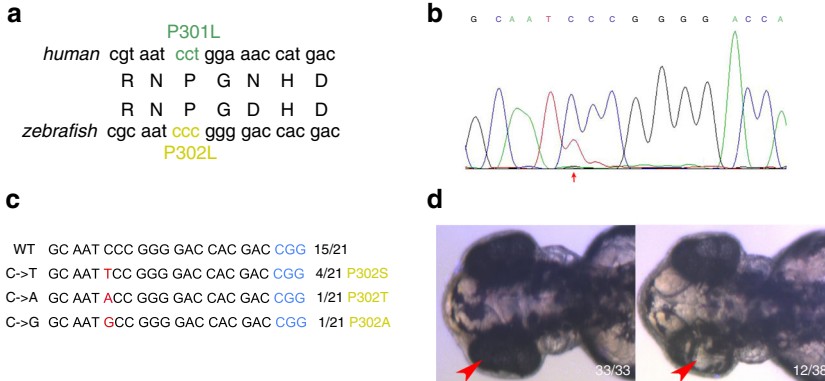

**Fig. 2** Induction of amino-acid change at the *tyr* site in zebrafish. **a** The diagram of mutation of human OCA. **b** Sequencing results at the *tyr* target. **c** Amino-acid change at the *tyr* site. **d** Phenotypes of the injected *tyr* +/− zebrafish. *Red arrows* indicate the overlapped peaks. The substituted base and amino acid are marked with *red* and *yellow*, respectively. *Red arrowhead* indicates the deficient eye

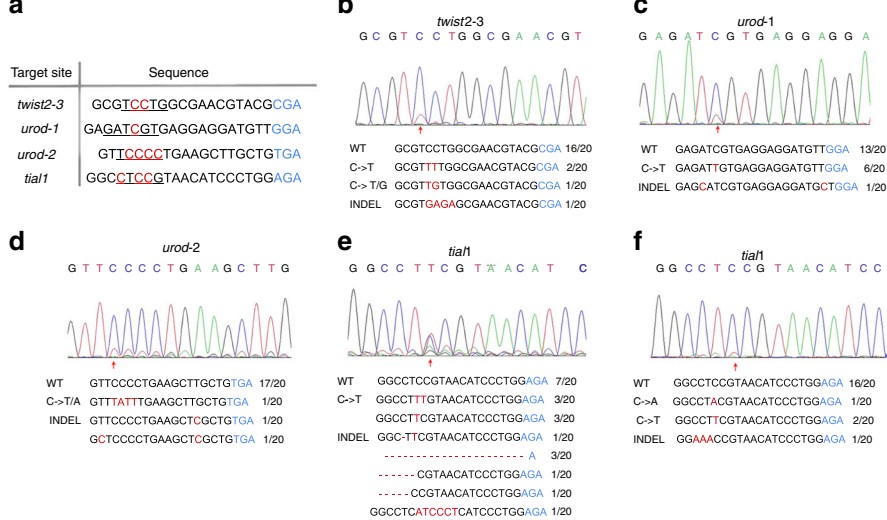

**Fig. 3** Base editing by Cas9-VQR variant. **a** The target site sequences. Target sequence (*black*), PAM region (*blue*), target sites (*red*), and target windows (*underlined*) are indicated. **b–e** Sequencing results of the *twist2–3*, *urod-1*, *urod-2*, and *tial1* targets. **f** Sequencing result of *tial1* site when using dCas9-VQR. *Red arrows* indicate the overlapped peaks. The substituted bases are marked with *red*. *Red dashes* represent the deleted bases in the sequence

sites (Fig. 3a). PCR sequencing results demonstrated that all targeted loci produced overlapping signal peaks at the base conversion site (Fig. 3b–e). It was noted that *urod* site2 and *tial1* had some unexpected lower overlapping signal peaks at other bases apart from the cytidine site (Fig. 3d, e), suggesting indels formation. The BE events at these four VQR sites were confirmed by sequencing colonies (Fig. 3b–e). Collectively, these results demonstrate that BE-VQR/gRNA system is also functional in zebrafish.

Interestingly, the high base conversion rate of *tial1* site was accompanied with high indel formation: 7 out of 20 colonies carried indels, while 6 out of 20 carried base substitutions (Fig. 3e). Since it was reported previously that replacement of nCas9 in BE by dCas9 would reduce the indel formation rate[8], we mutated the nCas9-VQR in BE-VQR to dCas9-VQR and assessed the base conversion rate at the same *tial1* site. As shown by the sequencing data (Fig. 3f), the base conversion rate dropped from 30 to 15% compared to the BE-VQR injected group. At the same time the indel formation rate also dropped from 35 to 5%. This is consistent with the previous report that using dCas9 decreases indel formation at the expense of lower base conversion efficiency[8].

## Discussion

To the best of our knowledge, this is the first demonstration that specific base conversion with high efficiency can be achieved in zebrafish without the need of template DNA. This finding further expands the application of the CRISPR-Cas9 system in zebrafish. Consistent with previous results in cultured cells, our results demonstrate that the BE system can induce cytidine to thymine conversion at specific sites in zebrafish. At the same time, substitution of cytidine to adenine or guanine can also be achieved. The germline frequency of precise modification at the four sites by BE system is 7–37%. This is significantly better than the reported rate of 4% for HDR[18], suggesting this BE system can be an efficient strategy for single base editing in zebrafish. In the case of BE-VQR system, we detected slightly higher indel rates in zebrafish, but silencing both nuclease domains of Cas9 reduced the indel formation. And recently, this system has also been reported to work in human cells, with efficiency of up to 50%[19]. The engineered Cas9-VQR variant with 5′-NGA PAM specificities expands the application of this BE system in zebrafish, providing an additional base editor.

For diseases caused by dominant mutations, simple knockout of the gene cannot mimic the function gained by the mutation. In such case, only introduction of the exact same mutation can reproduce the altered protein. Furthermore, many diseases have been shown to be heterogeneous, like OCA. Multiple point mutations of the same gene are found in patients with these diseases[14]. The BE system should be very useful in addressing what mutations are causing factors of the disease, as demonstrated by the *tyr* p.P301L locus described in our study.

Recently, another BE system using dCas9 fused with PmCDA1 from sea lamprey has been reported to correct point mutations[9, 20, 21]. Unlike the rAPOBEC1-nCas9-based BE system, the 3–5 bases surrounding the −18 position upstream of PAM sequence are frequently subjected to mutation[20]. We tested this PmCDA1-nCas9-based BE system in zebrafish and achieved positive base editing results at lower frequency. Given that zebrafish codon-optimized Cas9 improved the BE system in this study, we speculate that it might be helpful to use a zebrafish cytidine deaminase homologs to replace the rat APOBEC1[22]. However, that is outside the scope of this paper.

Taken together, this BE system is effective in inducing base conversion in zebrafish in a programmable manner. It provides a simple and efficient method for generating zebrafish models that can precisely mimic mutations found in human diseases.

## Methods

**Zebrafish husbandry**. Wild-type AB zebrafish and *tyr* +/− fish were raised and maintained at 28.5 °C in circulating system. All zebrafish experiments were approved by Institutional Animal Care and Use Committee (IACUC) of Peking University. The reference from IACUC of Peking University is LSC-ZhangB-1.

**BE-gRNA or BE-VQR-gRNA design**. The original rAPOBEC1-XTEN-Cas9n-UGI-NLS plasmid was obtained from Addgene (#73021)[8]. The nCas9 was replaced by a zebrafish codon-optimized version, and a BamH1 recognition site was inserted after the polyA sequence for the purpose of linearization. The D1135V/R1335Q/T1337R mutation of Cas9 was performed on BE plasmid to obtain the BE-VQR. And the H840A mutation of Cas9 was introduced into BE-VQR to obtain dBE-VQR. All the target sites are listed in Supplementary Table 1.

**In vitro synthesis of capped mRNA and gRNA**. The BE, BE-VQR, and dBE-VQR plasmid was linearized by BamH1, and capped mRNA was synthesized using Ambion mMESSAGE mMACHINE kit. The gRNA template was prepared according to the cloning-independent gRNA generation method[23]. And the primers are listed in Supplementary Table 2. The gRNA was synthesized using Ambion MAXIscript kit (Ambion) and purified using an RNeasy FFPE kit (Qiagen).

**Zebrafish microinjection and genotyping**. About 2 nl mixture of BE mRNA (300 ng μl⁻¹) and gRNA (25 ng μl⁻¹) was co-injected into one-cell stage wild-type embryos. Injected embryos were incubated at 28.5 °C for examination of phenotypes. After 2 dpf, embryos that developed normally were collected for genotyping. Genomic DNA was extracted from pools of six randomly collected embryos by alkaline lysis buffer-based DNA extraction. The genomic region surrounding the target site for each gene was PCR amplified and cloned into the pEASY-T1 vector (Transgene). Both the PCR fragments and the colonies were sequenced. Genotyping was performed for all the embryos injected with BE/ *tyr* gRNA individually. All the primers for detection are listed in Supplementary Table 2. Other injected embryos were raised to F0 adult, and then crossed with wild-type fish. The F1 embryos were collected in three groups, seven embryos each, and genomic DNA was extracted for each group. The fragments covering the target site were amplified and sequenced. F0 founders could be identified from the sequencing result. Each experiment was repeated for three times.

**Imaging**. Zebrafish embryos were anesthetized with 0.03% Tricaine (Sigma-Aldrich), and mounted in 4% methylcellulose. Photographs were taken by a Zeiss Axio Imager Z1 microscope, and processed by Adobe Photoshop CS6software.

**Data availability**. The authors state that all data necessary for confirming the conclusions presented in the article are represented fully within the article or available from the authors upon request.

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

## Acknowledgements

We thank Zenghou Tang for zebrafish husbandry. This work was supported in part by funding from Science and Technology Program of Shenzhen (JCYJ20150924110425180 and JCYJ20151030170755264) and NIH (R21GM109908). We thank Jenny Lin for editing the manuscript.

## Author contributions

Y.Z., W.Q., So.L., and S.L.: Conceived and designed experiments; Y.Z., W.Q., X.L., J.S., H.H., and H.B.: Performed the cloning, RNA synthesis, injection, and genotyping; W.Q.: Performed the imaging. Y.Z., W.Q., and S.L.: Wrote the manuscript.

## Additional information

**Competing interests:** The authors declare no competing financial interests.

