## [Peer Review file · Nature Communications]

Reviewers' comments:

Reviewer #1 (Remarks to the Author):

The authors' primary conclusion that base editing can be utilized *in vivo* in zebrafish is nicely supported by their data showing editing at 8 different genomic loci.

MAJOR ISSUE:

The major missing experiment that must be added prior to publication is a comparison with Cas9 nuclease with a donor DNA (HDR template). This paper is fundamentally a methods paper, and it is therefore critical that the authors compare their method with the current state-of-the-art methodology to achieve precise mutations in zebrafish, namely Cas9 nuclease + a donor HDR template. The authors should be able to do this comparison for several sites in a relatively straightforward manner, and the results are needed to support their claims that the BE system should work more efficiently and with less indels than HDR approaches.

If this major omission is addressed, then the main finding - that the authors were able to do base editing in zebrafish - is significant enough to the zebrafish and genome editing fields to make this paper deserving of publication in *Nature Communications*. The inclusion of two human disease-relevant SNP examples highlights the utility of this study. Assuming the comparison with Cas9 nuclease and a HDR donor template shows that the BE method works better in zebrafish, then the methods described in this paper will likely be used to more easily develop models of human genetic disease in these animals.

EDITORIAL ISSUES:

The statement "the engineered Cas9 VQR variant with altered PAM specificities expands the application of this BE system, providing additional base editors" is not true and is misleading. This variant has ONE altered PAM specificity, and whereas the sentence implies the paper has generated base editors with multiple newly accessible PAMs. The authors use this misleading language in their abstract as well, where they state "In addition, an engineered Cas9 VQR variant with altered PAM specificities induced base conversion in zebrafish, expanding the utility of this technology." They should specifically state that this new variant has an NGA PAM rather than say "altered PAM specificities"; indeed, this phrase looks as though it was copied directly from reference 12, which describes several new PAM variants of Cas9, rather than only one.

On a related note, the manuscript needs to be edited by a native English speaker, as there are serious grammatical mistakes throughout the entire manuscript that make the paper difficult to follow at times.

Overall, I recommend revision to include the key experiment described above, then reconsideration of the resulting revised manuscript.

Reviewer #2 (Remarks to the Author):

In the manuscript entitled "Programmable base editing of zebrafish genome using a modified CRISPR/Cas9 system", Lin and colleagues use modified CRISPR/Cas9 system to introduce C to T mutations in zebra fish. This base editing (BE) system utilizes the ability of gRNA and nuclease defective Cas9 to target APOBEC to defined genomic targets and employs Ugi to inhibit UNG-mediated repair of resultant G:U mismatches. The authors show that the system works for several test genes including *tiwst2*, *gdf6*, *ntl* and *tyr*. They then modify the BE system by using Cas9-VQR variants to expand target sequence beyond the PAM recognition motif. Overall, the results highlight an efficient and easy to manipulate system for base editing in zebra fish.

There have been many papers describing modifications of CRISPR/cas9 in manipulating the genome, including in zebrafish. The Apobec-dependent base modification has also been described extensively. Thus, this manuscript lacks novelty, yet using the Ugi system to prevent BER could be a convenient advance, if additional experiments are done to bolster the applicability of the system.

1. Can the effect of using Ugi in the system be quantified?
2. Can other APOBEC/AID family members be used to change the specificity of the target sequence?
3. UNG and the BER pathway allows error-free repair and using Ugi might increase general mutations. How can this be scored or evaluated?

Reviewer #3 (Remarks to the Author):

The manuscript by Zhang et al, "Programmable base editing of zebrafish genome using a modified CRISPR/Cas9 system", demonstrates efficient base-editing by modified BE systems in zebrafish. This is the first successful application of the BE system in vivo. The system seems to be more efficient than the published methods using templates with long or short homology arms. The paper should be of broad interest to zebrafish researchers as well as researchers of other model organisms. The availability of these reagents will likely accelerate disease modeling in zebrafish. There are some concerns that need to be addressed:

Major concerns.

- 1) Although the focus of this manuscript is to demonstrate that BE works in vivo, which is not all that surprising, the characterization of the phenotypes from the mutations generated should enhance the significance of this work. For example, does the E78K allele of twist2 cause developmental defects akin to Ablepharon macrostomia syndrome (AMS)?
- 2) The conclusion that P301 alleles of tyr are responsible for the observed ocular and oculocutaneous albinism (OCA) needs to be strengthened. The authors did not state the frequency of phenotypic embryos. It could be a result of rare indels. Confirmation from germline mutations should make it unambiguous.
- 3) The editing efficiencies determined in the manuscript likely represent somatic cells. It would be more informative for readers to know the germline transmission rate of the correctly edited alleles from the mutagenized founders.

Minor issues:

- 1) There are a few cases of imprecise word usage. For example: line 55, "double strand break" may be better than "DNA cleavage"; line 128, "requirement" may be better than "recognition"; line 157, "both" may be better than "all"; and line 175 "in" maybe better than "of".
- 2) For the sake of completeness, please cite other base editor systems in the sentence in line 177, including PMID: 27723754 and PMID: 27798611.
- 3) Line 93, the trace of ntl (Fig 1d) seems to suggest more indel near the PAM than others. Maybe more colonies need to be sequenced, particularly for the low efficiency cases?

Response to Reviewers

We thank the reviewers and editor for constructive critiques. We revised the manuscript accordingly. We think the paper is improved and hope it is acceptable for publication in Nature Communication. A copy with track changes is also included.

Comments from reviewer(s):

Reviewer #1 (Comments to the Author):

MAJOR ISSUE:

The major missing experiment that must be added prior to publication is a comparison with Cas9 nuclease with a donor DNA (HDR template). This paper is fundamentally a methods paper, and it is therefore critical that the authors compare their method with the current state-of-the-art methodology to achieve precise mutations in zebrafish, namely Cas9 nuclease + a donor HDR template. The authors should be able to do this comparison for several sites in a relatively straightforward manner, and the results are needed to support their claims that the BE system should work more efficiently and with less indels than HDR approaches.

Response: HDR templates can be long double stranded DNA or single stranded oligo. To use long double stranded DNA templates, based on our previous experience, isogenic zebrafish are needed to prepare the donor DNA fragments with true sequence homology. This is labor intensive and time consuming. For single base editing, ssODN mediated HDR has been reported to be effective. Here in revision, we therefore designed four ssODN to compare efficiency with the BE system at the same sites. To our surprise, we could not detect any precise modification by ssODN method at these sites. What we obtained was modification with sequence errors at 3' end of the targeting site (Fig S1 here for reviewers). We don't feel that we can conclude BE system is much better than ssODN based on our data from this small sample size. Instead, we present new data of germline transmission frequency of BE system. The germline frequency of precise modification at the four sites by BE system is 7-37%, while frequency of published single base modification is less than 4%(Albadri et al. 2017).

Fig S1 The HDR integration results mediated by ssODN template for four sites in zebrafish. For *twist2-1* (A) and *ntl* sites (B), NheI and EcoRI restriction site was introduced in the template individually. For *twist2-2* (C) and *tyr* sites (D) uncut bands (red arrowhead) after BsrBI or XmaI digestion were extracted for further analysis. (E) The sequencing result confirmed the imprecise integration event at *ntl* site. The five target point mutations were labeled by red color.

EDITORIAL ISSUES:

The statement “the engineered Cas9 VQR variant with altered PAM specificities expands the application of this BE system, providing additional base editors” is not true and is misleading. This variant has ONE altered PAM specificity, and whereas the sentence implies the paper has generated base editors with multiple newly accessible PAMs. The authors use this misleading language in their abstract as well, where they state “In addition, an engineered Cas9 VQR variant with altered PAM specificities induced base conversion in zebrafish, expanding the utility of this technology.” They should specifically state that this new variant has an NGA PAM rather than say “altered PAM specificities”; indeed, this phrase looks as though it was copied directly from reference 12, which describes several new PAM variants of Cas9, rather than only one.

Response: As suggested, the sentence in abstract now reads: “In addition, an engineered Cas9 VQR variant with 5'-NAG PAM specificities was used to induce base conversion in zebrafish. This shows that Cas9 variants can be used to expand the utility of this technology.” The sentence in discussion now reads: “The engineered Cas9 VQR variant with 5'-NAG PAM specificities expands the application of this BE system, providing an

additional base editor.”

On a related note, the manuscript needs to be edited by a native English speaker, as there are serious grammatical mistakes throughout the entire manuscript that make the paper difficult to follow at times.

Response: Done. A native English speaker has edited the manuscript.

Reviewer #2

There have been many papers describing modifications of CRISPR/cas9 in manipulating the genome, including in zebrafish. The Apobec-dependent base modification has also been described extensively. Thus, this manuscript lacks novelty, yet using the Ugi system to prevent BER could be a convenient advance, if additional experiments are done to bolster the applicability of the system.

1. Can the effect of using Ugi in the system be quantified?

Response: To address this issue, we made a construct with UGI domain deleted. At the three sites tested in zebrafish, we no longer were able to detect base modification (Fig S2 here for reviewers), suggesting low efficiency of the BE system without UGI.

Fig S2. After removing the UGI domain in the BE system, the single base conversion cannot be detected at three sites in zebrafish. The potential deamination sites are indicated by red lines.

2. Can other APOBEC/AID family members be used to change the specificity of the target sequence?

Response: Recently, another study using dCas9 fused with PmCDA1 from sea lamprey has been reported to correct the point mutation. We tested this system at the same sites. Since we observed lower base editing efficiency of this system in zebrafish, we didn't do further work on it.

3. UNG and the BER pathway allows error-free repair and using Ugi might increase general mutations. How can this be scored or evaluated?

Response: Since UGI is essential for BE system to work in zebrafish, we have to keep it. The potential problem of increasing indels by UGI can be solved by screening germline transmission of modification in individual fish. In revision, we added data of frequency of germline transmission of four sites with precise modification, which range from 7 to 37%. Fish with unwanted indels can be discarded.

Reviewer #3

The manuscript by Zhang et al, "Programmable base editing of zebrafish genome using a modified CRISPR/Cas9 system", demonstrates efficient base-editing by modified BE systems in zebrafish. This is the first successful application of the BE system in vivo. The system seems to be more efficient than the published methods using templates with long or short homology arms. The paper should be of broad interest to zebrafish researchers as well as researchers of other model organisms. The availability of these reagents will likely accelerate disease modeling in zebrafish. There are some concerns that need to be addressed:

Major concerns.

1) Although the focus of this manuscript is to demonstrate that BE works in vivo, which is not all that surprising, the characterization of the phenotypes from the mutations generated should enhance the significance of this work. For example, does the E78K allele of twist2 cause developmental defects akin to Ablepharon macrostomia syndrome (AMS)?

Response: We are just getting germline transmission of this mutation and need more time to determine if AMS phenotype will develop. With high frequency of germline transmission

of all expected modifications obtained, we feel the method itself is significant to be published. We wish this reviewer would allow us to address the phenotype in future studies.

2) The conclusion that P301 alleles of tyr are responsible for the observed ocular and oculocutaneous albinism (OCA) needs to be strengthened. The authors did not state the frequency of phenotypic embryos. It could be a result of rare indels. Confirmation from germline mutations should make it unambiguous.

Response: We agreed with this critique. It turns out that P301 (P302 in zebrafish) alleles are more complicated than we thought since there are three Cs at the targeting site that can be changed to generate multiple alleles (we even found another new change of CCC to TTC in germline). We need germline transmission of each individual mutation to determine which one would induce OCA. In revision, we added that the current observation of OCA phenotype in injected F0 embryos could be a result of indels. Again, we wish to address P302 germline phenotype issue in future studies by breeding individual germline mutation to homozygosity. The frequency of phenotypic embryos in Figure 2 was added.

3) The editing efficiencies determined in the manuscript likely represent somatic cells. It would be more informative for readers to know the germline transmission rate of the correctly edited alleles from the mutagenized founders.

Response: Data of germline transmission of 7-37% has been added in the revised manuscript.

Minor issues:

1) There are a few cases of imprecise word usage. For example: line 55, "double strand break" may be better than "DNA cleavage"; line 128, "requirement" may be better than "recognition"; line 157, "both" may be better than "all"; and line 175 "in" maybe better than "of".

Response: As suggested, we revised these words.

2) For the sake of completeness, please cite other base editor systems in the sentence in line 177, including PMID: 27723754 and PMID: 27798611.

Response: Done. We cited these references.

3) Line 93, the trace of *ntl* (Fig 1d) seems to suggest more indel near the PAM than others. Maybe more colonies need to be sequenced, particularly for the low efficiency cases?

Response: Done. We sequenced more colonies for *ntl* site and did not find more indels at this region.

1. Albadri, S., F. Del Bene and C. Revenu (2017). Genome editing using CRISPR/Cas9-based knock-in approaches in zebrafish. *Methods*.

REVIEWERS' COMMENTS:

Reviewer #1 (Remarks to the Author):

The authors should cite the recent Kim et al. Nature Biotechnology paper reporting several new base editors including the VQR editor described in this manuscript. I don't think the Kim paper should preclude the publication of this manuscript since additional observations in this work are significant contributions to the field, but Kim should be cited as it is partially overlapping.

Other recent papers (by Jin-Soo Kim and others) have been published demonstrated in vivo base editing, so the authors should also remove claims suggesting that this work is the first example of in vivo base editing since such claims are incorrect.

Beyond these two important issues to be corrected, the authors have satisfactorily addressed the concerns previously raised by the reviewers in my opinion.

Reviewer #2 (Remarks to the Author):

The authors have more or less addressed the issues raised by the reviewer

Reviewer #3 (Remarks to the Author):

I am satisfied with the revised manuscript. I recommend acceptance for publication.

Comments from reviewer(s):

Reviewer #1 (Remarks to the Author):

The authors should cite the recent Kim et al. Nature Biotechnology paper reporting several new base editors including the VQR editor described in this manuscript. I don't think the Kim paper should preclude the publication of this manuscript since additional observations in this work are significant contributions to the field, but Kim should be cited as it is partially overlapping.

Response: Done. We added the sentence “And recently, this system has also been reported to work in human cells, with efficiency of up to 50%¹⁹” in the discussion and cited the reference.

Other recent papers (by Jin-Soo Kim and others) have been published demonstrated in vivo base editing, so the authors should also remove claims suggesting that this work is the first example of in vivo base editing since such claims are incorrect.

Response: Done. We added the sentence “To date, this system has been reported to work in mouse and several crops⁹⁻¹¹” in the introduction and cited those references.

Beyond these two important issues to be corrected, the authors have satisfactorily addressed the concerns previously raised by the reviewers in my opinion.

Reviewer #2 (Remarks to the Author):

The authors have more or less addressed the issues raised by the reviewer

Reviewer #3 (Remarks to the Author):

I am satisfied with the revised manuscript. I recommend acceptance for publication.